# Photovoltaic Properties of ZnO Films Co-Doped with Mn and La to Enhance Solar Cell Efficiency

**DOI:** 10.3390/nano12071057

**Published:** 2022-03-24

**Authors:** Muhammad Amjad, Muhammad Iftikhar Khan, Norah Alwadai, Muhammad Irfan, Hind Albalawi, Aljawhara H. Almuqrin, Maha M. Almoneef, Munawar Iqbal

**Affiliations:** 1Department of Physics, The University of Lahore, Lahore 53700, Pakistan; amjad.karyal@gmail.com (M.A.); muhammad.irfan6252@gmail.com (M.I.); ikrmhaq@gmail.com (I.-u.-H.); 2Department of Physics, College of Science, Princess Nourah bint Abdulrahman University, P.O. Box 84428, Riyadh 11671, Saudi Arabia; hmalbalawi@pnu.edu.sa (H.A.); ahalmoqren@pnu.edu.sa (A.H.A.); mmalmoneef@pnu.edu.sa (M.M.A.); 3Department of Chemistry, The University of Lahore, Lahore 53700, Pakistan

**Keywords:** ZnO, co-doping, Mn and La, dye-sensitized solar cells, structural, optical properties

## Abstract

In the present investigation, ZnO films co-doped with Mn and La were synthesized by the sol–gel technique. XRD analysis revealed that ZnO had a hexagonal structure. Mixed hexagonal and cubic phases appeared in ZnO containing Mn (1%) and La (1.5%). The grain size, d-spacing, unit cell, lattice parameters, atomic packing fraction, volume, strain, crystallinity, and bond length of co-doped ZnO films were determined as a function of doped ion contents. Through UV analysis, it was found that pristine ZnO had E_g_ = 3.5 eV, and it decreased when increasing the doping concentration, reaching the minimum value for the sample with 1% Mn and 1% La. The optical parameters of the films, such as absorption, transmittance, dielectric constants, and refractive index, were also analyzed. DSSCs were fabricated using the prepared ZnO films. For pure ZnO film, the values were: efficiency = 0.69%, current density = 2.5 mAcm^−2^, and open-circuit voltage = 0.56 V. When ZnO was co-doped with Mn and La, the efficiency increased significantly. DSSCs with a ZnO photoanode co-doped with 1% Mn and 1% La exhibited maximum values of J_sc_ = 4.28 mAcm^−2^, V_oc_ = 0.6 V, and efficiency = 1.89%, which is 174% better than pristine ZnO-based DSSCs. This material is good for the electrode of perovskite solar cells.

## 1. Introduction

Due to its economic benefits, solar energy is used as a clean and cheap renewable energy source. Significant factors that increase solar energy demand are environmental crises and the large global consumption of energy to support human activities. The sun is the most abundant source of renewable energy [1]. Solar cells directly convert sun energy into electricity, which requires a solar device. Nanotechnology is a viable approach to achieving this goal by using nanomaterials that are active under solar light to harvest energy for different purposes [2]. Dye-sensitized solar cells (DSSCs) are easier to fabricate compared to Si-based solar cells and, therefore, are viable alternatives [3,4,5,6,7]. Low manufacturing costs, high interest, and human-friendly environmental benefits are key advantages of DSSCs [8,9]. DSSCs have four fundamental parts: a photoanode, counter electrode, dye, and electrolyte. When light falls on the cell surface, photons from sunlight enter the cell. Then, holes move towards the highest occupied molecular orbital (HOMO), while electrons move towards the dye’s lowest unoccupied molecular orbital (LUMO). Because the LUMO level of the dye is higher than the conduction band of TiO_2_, electrons from the LUMO level move towards the conduction band of TiO_2_. The efficiency of solar cells is affected by these transferred electrons. Ye N. et al. [3] proposed that the number of transferred electrons is proportional to the photocurrent. Therefore, among the aforementioned components, the photoanode is the most critical component of the cell. It absorbs dye and accepts electrons from the dye. Consequently, it must have a porous structure. Because of its porous structure and wide band gap, TiO_2_ is considered the best candidate as a photoanode for DSSCs. Comprehensive studies have been carried out on the use of TiO_2_ NPs as a photoanode since 1991 [10,11]. TiO_2_ nanoparticle films, when used as a photoanode, show more than 14% light conversion efficiency due to a highly efficient surface area. However, low electron mobility and poor long-term stability are still significant challenges [12]. Studies have been conducted on other alternative semiconductor oxides, such as ZnO, SnO_2_, Zn_2_SnO_4_, and Nb_2_O_5_ [13,14,15,16]. Among these semiconductors, ZnO has been found to be a promising alternative photoanode for DSSCs. Because its E_g_ and electron affinity are close, 3.3 eV and 3.2 eV, respectively, ZnO possesses excellent electron diffusivity, which is higher than that of TiO_2_ [17]. At the same time, ZnO is a material that is rich in diverse morphologies when compared with any other material [18]. Additionally, ZnO is a low-cost material, is resistant to photocorrosion, has a high excitation binding energy (60 MeV), and has a surface that is easier to modify [19]. However, DSSCs with ZnO-based photoelectrodes show low photovoltaic performance due to the following reasons: (1) they absorb light in a narrow spectral range, (2) the recombination rate is fast, and (3) the film is unstable against acidic dye [20]. In principle, the aforementioned drawbacks are ignored if electrodes based on ZnO can attain a comparable or even superior photovoltaic performance [21]. One of the possible ways to overcome the above-mentioned drawbacks is doping with foreign metals such as Cu, Mn, Au, La, and Ni. In this research work, Mn and La have been doped into ZnO. Both Mn and La can be replaced with Zn^2+^ ions, and they can be inserted into the lattice of ZnO. Additionally, when transition metals are doped into ZnO, new energy levels are formed because of the d-orbitals of the transition metals. These new energy levels can enhance the electrical and optical properties of ZnO by changing the band gap. Moreover, the open-circuit voltage and efficiency of the solar cell are affected by adsorption and dye injection electrons on the ZnO surface [22]. Furthermore, in the literature, it is predicted that when Zn^2+^ ions or oxygen vacancies are doped with metal, the metal is able to promote the binding of dye molecules to the ZnO surface [23]. Different transition metals have been doped into ZnO, such as Mn, Co, Cu, and La [14]. Mn has different features when doped into ZnO, such as a large moment, a tiny difference in ionic radius between Zn^2+^ and Mn^2+^, and a nearly half-filled 3*d* orbital. Because of these features, Mn is incorporated into the ZnO lattice for DSSCs. ZnO nanoparticles are important host material for La ions, because they are a well-known material for providing luminescence in doped particles. Therefore, co-doping ZnO with Mn and La could overcome the above-mentioned drawbacks and enhance the cell efficiency [24,25].

In this work, films of ZnO, 1% Mn-ZnO, and ZnO co-doped with 1% Mn and 0.5%, 1%, or 1.5% La were fabricated by the sol–gel technique, which is a suitable technique for easy and large-area deposition for industrial purposes. Many structural properties, such as the phases of ZnO, grain size, d-spacing, lattice parameters, atomic packing fraction, strain, crystallinity, volume of the unit cell, and bond length, were calculated. Optical parameters such as band gap, absorption, refractive index, transmittance, and dielectric constants were calculated. After that, DSSCs were prepared by using these films, and their efficiency was compared. To the best of our knowledge, this is the first study in the literature to conduct the above-described analyses. This work provides a new way of co-doping to improve the optical, electrical, and photovoltaic properties of different materials.

## 2. Experimentation

Solution 1 was prepared by stirring 2 g of zinc acetate (Sigma-Aldrich, Gillingham, UK) in 15 mL of distilled water for 10 min. Solution 2 was made by dissolving 8 g of sodium hydroxide (Sigma-Aldrich, Gillingham, UK) into 10 mL of H_2_O and stirred for 5 min to obtain homogeneity. Both solutions 1 and 2 were mixed by stirring for 1 h to obtain the ZnO gel. The chemical reaction in this solution is given below.
Zn (CH_3_ COO)_2_·2H_2_O + 2NaOH → ZnO + NaCH_3_COO + 2H_2_O

To prepare ZnO co-doped with 1 wt.% Mn and 0 wt.%, 0.5 wt.%, 1 wt.%, and 1.5 wt.% La, 1 wt.% manganese nitrate and lanthanum nitrate (Sigma-Aldrich, UK) were added to solution 1, and the same steps were repeated. In this manner, five solutions were prepared: undoped ZnO and ZnO co-doped with 1% Mn and 0%, 0.5%, 1%, and 1.5% La. Five pieces of fluorene-doped tin oxide (FTO)-glass were ultrasonically washed in acetone, ethanol, and DI water for 17 min in turn. After drying the substrates, films of ZnO were coated on FTO-glass substrates by the dip-coating method and heated at 120 °C for 10 min. Doped films with other compositions were coated with the same procedure and heated at 120 °C for 10 min. Finally, all films were calcinated at 300 °C for 2 h. 

The structures of pure and co-doped ZnO were studied by X-ray diffraction (XRD) (PANalytical X’Pert PRO). Optical HITACHI U-2800 UV–vis spectrometer was used to investigate optical characteristics. 

For the fabrication of solar cells, films were sensitized in a 0.05 mM ethanol solution of N719 dye for 22 h. Weakly adsorbed dye was rinsed off with absolute ethanol, and the films were dried in air and then covered with the platinum-coated FTO-glass substrate as the counter electrode. An electrolyte of tri-iodide and iodide solution filled the space between the photoanode and counter electrode for the regeneration of electrons, and then both plates were sealed. 

DSSC J-V characteristics were measured using a solar simulator (AM 1.5, 100 mW c/m^2^, Newport, Keithley 2400) was used [26]. The thickness of the film was measured using a surface profilometer. 

## 3. Results and Discussion

### 3.1. XRD

The structure of the crystalline material, phases, and unit cell dimensions were investigated by XRD. XRD patterns for pure ZnO and ZnO films co-doped with 1% Mn and 0%, 0.5%, 1%, and 1.5% La are presented in Figure 1. Figure 1 shows that for the pure ZnO film, two diffraction peaks occurred at 2θ angles of 36.19° and 55.93°, corresponding to (101) and (110) crystal planes of the hexagonal phase structure; the absence of any other contamination peaks confirms the purity of ZnO material, according to PDF no. 36-1451 [27]. When 1% Mn was doped into ZnO, the same two diffraction peaks were detected at 2θ angles of 36.11° and 55.99°, corresponding to the (101) and (110) planes of the ZnO hexagonal phase structure. The intensities of these peaks are slightly increased, and the peaks are shifted to lower angles as compared to the pure ZnO film. As other peaks, such as MnO and Mn-Zn, are not observed, this confirms the successful doping of Mn [28]. Mn was successfully absorbed into ZnO as substitutional atoms, as evidenced by the peak shifts to lower angles [29,30]. With 0.5% La doped into 1% Mn-doped ZnO film, the same two diffraction peaks occurred at 2θ angles of 36.05° and 55.93°, corresponding to (101) and (110) crystal planes of the hexagonal phase structure. Here, an improvement in peak intensity and small angle shifts are observed, confirming the successful doping of La. When the La doping concentration increased from 0.5% to 1% in 1% Mn-doped ZnO, two diffraction peaks occurred at 2θ angles of 36.00° and 55.88°, corresponding to (101) and (110) planes. The improvement in intensity and peak shifts indicate that La^3+^ and Mn^2+^ were uniformly dispersed in ZnO. The undoped ZnO and ZnO films co-doped with Mn and La exhibit crystalline behavior, which is indicated by sharp diffraction peaks [31]. La did not disturb the hexagonal structure of ZnO when it was doped into ZnO samples, indicating that La was uniformly dispersed over Zn^2+^ [32]. The (101) plane shows maximum growth in each sample, which is the preferred orientation in each sample. The information obtained is consistent with JCPDS card no. 36–1451 and shows a hexagonal wurtzite structure.

When the La doping concentration was increased to 1.5%, three diffraction peaks occurred at 2θ angles of 35.93°, 55.88°, and 66.99°, corresponding to (101), (110), and (311) crystal planes. The first two peaks correspond to (101) and (110) and indicate a hexagonal structure (card no. 36-1451), but the peak corresponding to (311) reveals a cubic phase structure of ZnO. When 1.5% La is doped into 1% Mn-ZnO, La atoms cannot be accommodated in the interstitial spaces but disperse and arrange into a cubic structure, so a change in the phase of the material can be observed [33]. The variation in the phase may be attributed to the greater atomic size of La compared to that of Zn, which may affect the vibration of the atoms [34,35]. 

#### 3.1.1. Grain Size

Grain size is calculated by the Debye–Scherrer formula [36].
(1)$$D=\frac{0.94\;\lambda }{\beta \cos\theta }$$

In Equation (1), D is the grain size measured in nanometers, and λ is the X-ray wavelength, the value of which is 0.154 nm. β is the full width at half maximum (FWHM) of the diffraction peak, and θ is Bragg’s diffraction angle. The grain sizes calculated from the intense (101) peaks for pure ZnO and ZnO films doped with 1% Mn and 0%, 0.5%, 1%, and 1.5% La are 13 nm, 16 nm, 20 nm, 27 nm, and 22 nm respectively.

Figure 2 shows that the grain size for pure ZnO is 13 nm. When 1% Mn was doped into ZnO, the grain size increased to 16 nm. These increasing grain sizes show that the orientations of atoms are well arranged compared to pure ZnO. The atomic radius of Mn is greater than that of Zn; therefore, stress is generated in the bonds of the material, which is responsible for this increase in grain size [37]. When doping 0.5% and 1% La into 1% Mn-ZnO, the values of the grain sizes are 20 nm and 27 nm, respectively. These values show that the atoms have better orientations compared to the previous samples. The atomic radius of La is slightly greater than that of Mn, which accounts for the increased grain size [38]. When 1.5% La was doped into 1% Mn-ZnO, the value of the grain size decreased. Due to the large quantity of La, the orientations of the atoms are not as well arranged as in the previous samples, which is responsible for this decrease in grain size [39].

#### 3.1.2. D-Spacing

The perpendicular distance between atom planes is known as d-spacing. For each peak, corresponding d-spacing was calculated from a diffractogram. Within the crystal, atoms in the planes can be described using a 3D coordinate system. Bragg’s law is used to calculate the d-spacing between the planes of the atoms in each sample [26].
(2)2dsinθ=nλ

The d-spacings of pure ZnO and ZnO co-doped with 1% Mn and 0%, 0.5%, 1%, and 1.5% La are 2.48 Å, 2.52 Å, 2.53 Å, 2.54 Å and 2.54 Å, respectively. These values were calculated for the (101) plane of each sample. A graph of the d-spacing of each sample is shown in Figure 3, which shows that the d-spacing value is slightly increased due to Mn and La doping. The large atomic radii of Mn and La cause this slight increase in d-spacing [40]. It might be possible that during the addition of impurities, La^3+^ ions fill the interstitial spaces. Therefore, the bonds between the atoms are slightly expanded; this explains the slight increase in d-spacing after doping [41].

#### 3.1.3. Lattice Parameter

Lattice parameters are the lengths along each crystallographic axis of a unit cell. Generally, there are three lattice constants, *a* = *b* and *c*. The lattice parameter for a hexagonal structure is calculated by the following formula [42].
(3)$$\frac{1}{d^{2}}=\frac{4}{3}\left(\frac{h^{2}+hk+k^{2}}{a^{2}}\right)+\frac{l^{2}}{c^{2}}$$

Miller indices and d-spacing between the layers of atoms of the hexagonal structure are represented by (hkl) and *d*, respectively. In Equation (3), *a* and *c* are the lattice parameters. The lattice parameters of pure ZnO and ZnO co-doped with 1% Mn and 0%, 0.5%, 1%, and 1.5% La were calculated by Equation (3), and the results closely coincide with standard values *a* = 3.24 Ǻ and *c* = 5.20 Ǻ as per the standard card (PDF no. 36-1451). 

Figure 4 shows the graph of parameters a and **c** versus samples. Parameter ‘*a*’ for pure ZnO is 3.28 nm, and parameter ‘*c*’ is 5.08 nm. When 1% Mn was doped into ZnO, the value of parameter ‘*a*’ decreased to 3.27 nm. The small decrease in parameter a is due to compression of the bonding. The value of ‘*c*’ is 5.59 nm. The shifting of the peak position to a lower angle when 1% Mn was doped into ZnO indicates the expansion of the ‘*c*’ parameter [43]. Deshmukh et al. [44] also observed a systematic increase in the lattice parameter when doping Mn into ZnO.

When 0.5%, 1%, or 1.5% La was doped into 1% Mn-ZnO, slight variations in ‘*a*’ and ‘*c*’ parameters were observed, confirming the successful doping of these materials into ZnO. Moreover, there was no significant difference between doped and pure lattice parameters, indicating the successful doping of Mn and La into ZnO. This variation in parameters affects the vibration of atoms and changes the optical and electrical characteristics of ZnO, affecting the efficiency of DSSCs.

#### 3.1.4. Packing Fraction and Strain

The volume fraction in a crystal structure occupied by constituent atoms is called the packing fraction [45]. The packing fraction for each sample is calculated by the following formula [46].
(4)$$APF=\frac{2\pi a}{3c\sqrt{\left(3\right)}}$$

The calculated values of the packing factor for pure ZnO and ZnO co-doped with 1% Mn and 0%, 0.5%, 1%, and 1.5% La are 78%, 71%, 73%, 71%, and 71%, respectively. It was found that the APF decreased with Mn and La doping in ZnO. This may be due to the decrease in voids in the samples [47]. This decrease in APF indicates that the volume occupied by atoms is increased because of the large atomic sizes of La and Mn compared to that of Zn. The Williamson–Hall (W-H) method is used to calculate the strain of each sample by the following formula [48].
(5)$$\varepsilon =\frac{\beta }{4\tan\theta }$$

For each sample, the strain values for pure ZnO and ZnO co-doped with 1% Mn and 0%, 0.5%, 1%, and 1.5% La are 0.008806, 0.008134, 0.007621, 0.007097, and 0.00707, respectively. The strain is observed to have a decreasing trend, showing a reduced flexibility tendency and indicating that the material has a packed structure.

#### 3.1.5. Crystallinity

The following equation was used to calculate the degree of the crystallinity ($X_{c}$) of ZnO co-doped with Mn and La [49].
(6)$$X_{c}=\frac{0.24}{\beta }$$

The values of the degree of the crystallinity ($X_{c}$) of pure ZnO and ZnO co-doped with 1% Mn and 0%, 0.5%, 1%, and 1.5% La were found to be 20.86 Ǻ, 22.60 Ǻ, 26.08 Ǻ, 24.24 Ǻ, and 26.08 Ǻ, respectively. This indicates the structure of the material is improved, resulting in better optical and electrical properties.

#### 3.1.6. Volume of Unit Cell, Bond Length, and Lattice Distortion

Bond length (*L*), volume of the unit cell (*V*), and lattice distortion (*R*) are calculated by the following formulas [50].
(7)$$V=\frac{\sqrt{3}}{2}a^{2}c$$
(8)$$L=\sqrt{\frac{a^{2}}{3}+{\left(\frac{1}{2}-u\right)}^{2}c^{2}}$$
where $u=\frac{a^{2}}{3c^{2}}+0.25$.
(9)$$R=\frac{2a\sqrt{2/3}}{c}$$

The ‘*V*’ of pure ZnO films and ZnO films co-doped with 1% Mn and 0%, 0.5%, 1%, and 1.5% La were found to be 47.27 Å^3^, 51.70 Å^3^, 50.81 Å^3^, 52.02 Å^3^, and 51.70 Å^3^, respectively [50]. The increase in the volume of the unit cell is appropriate for embedding and the conductivity of the material [51]. The values of the bond length (*L*) for the films are 1.97 Å, 2.03 Å, 2.02 Å, 2.03 Å, and 2.03 Å, respectively. The increase in bond length is due to the increase in ‘*V*’. The values of lattice distortion (*R*) for the films were found to be 0.61 Å, 0.55 Å, 0.56 Å, 0.55 Å, and 0.55 Å, respectively. The decrease in lattice distortion results in a reduction in the energy gap [52]. A summary of XRD parameters is presented in Table 1, which is obtained from Figure 1.

### 3.2. UV–Vis Analysis


Ultraviolet and visible (UV–vis) absorption spectroscopy was used to measure the optical properties of the films, such as absorbance, transmittance, optical band gap (*E_g_*), refractive index, excitation coefficient, and dielectric constants.

#### 3.2.1. Absorbance

The absorbance spectra were recorded using a UV–visible spectrophotometer to obtain more details about the dopant incorporation and the change in the E_g_ of pure and doped films. The UV–visible absorption spectra for pure ZnO and ZnO films co-doped with 1% Mn and 0%, 0.5%, 1%, and 1.5% La are presented in Figure 5.

Figure 5 reveals that the pure ZnO film absorbs in the UV region, indicating the formation of the ZnO film. With 1% Mn doping, the absorption edge is shifted towards a longer wavelength. This is due to the larger grain size, which is also evident in the XRD results. With La doping, a shift of absorption edges towards a longer wavelength is also observed. Various factors, such as particle size, oxygen deficiency, and grain structure defects, are responsible for the increase in absorbance [53]. XRD spectra also indicate an increase in grain size due to La doping. Therefore, absorbance edges are shifted towards a longer wavelength. This redshift is due to La doping, which is characterized by the s–d exchange interaction between the localized “d” electrons and the band electrons of the transition metal ion at the cationic site [54]. 

#### 3.2.2. Band Gap Calculation

Figure 6 shows the calculated values of *E_g_* for pure and co-doped ZnO films. *E_g_* is found by the following relation [55].
(10)$$\alpha hv=A{\left(hv-Eg\right)}^{n/2}$$
where *A* is a constant, *α* is the absorbance coefficient, and ‘*n*’ is a number indicating a direct or indirect band gap, i.e., *n* = 4 and 1 for direct and indirect *E_g_*. In the present work, its value is 4, which shows a direct band gap. 

Figure 6 is the plot of hν versus (αhν)^2^ for undoped and doped ZnO films. The *E_g_* for pure ZnO is 3.5 eV. With 1% Mn-ZnO, *E_g_* decreased to 3.4 eV. The drop in *E_g_* can be related to an increase in particle size and the concentration of oxygen vacancies on the ZnO surface [56]. Localized defect states within the band gap are created by the oxygen vacancy and the related disorder, resulting in a redshift of the band gap [57]. Additionally, a decrease in *E_g_* is attributed to the s–d exchange interactions between the localized and band electrons of Mn.

For 1% La and 1% Mn co-doped ZnO, *E_g_* decreased. When Mn and La are doped into ZnO, they create additional energy levels between the forbidden gaps of ZnO [58]. Due to these levels, an interaction between VB and CB occurs, which is responsible for this decrease in band gap. An increase in *E_g_* is observed at a doping concentration of 1.5% La in Mn-ZnO film. At this doping concentration, mixed phases of ZnO (i.e., hexagonal and cubic) form (Figure 1). This may be responsible for the increase in *E_g_*. It is worth noting that the direct optical band gap shifts to higher energy (blue shift) when increasing La concentrations. The widening (blue shift) of the energy band resulting from the increase in the Fermi level in the conduction band at lower doping levels is explained by the Moss–Burstein band-filling effect and quantum confinement effects [59].

#### 3.2.3. Transmittance of the Films

The transmittance spectra of various films are plotted in the wavelength range of 200 to 750 nm (Figure 7). For the photoanode of a DSSC, films must be highly transparent in the visible portion. The prepared ZnO films with co-doping levels of 1% Mn and 1% La are extremely transparent in the visible region (Figure 7), making them suitable for DSSCs. 

#### 3.2.4. Refractive Index

The refractive index (*n*) is the ratio of light speed in a vacuum to light speed in a substance. For optical materials, ‘*n*’ and its applications are important parameters. The value of ‘*n*’ is calculated with the use of reflectance (*R*) and the extinction coefficient (*K*) of the material. It is calculated by the expression in (11) [60].
(11)$$n=\sqrt{\frac{4R}{\left(1-R\right)2}-K^{2}}-\frac{R+1}{R-1}$$

In this equation, *R* and *k* are the reflectance and extinction coefficient. The extinction coefficient *K* is determined through the following relation [60].
(12)$$K=\frac{\alpha \lambda }{4\pi }$$
where α is the coefficient of absorption, calculated by α=2.303A/d. *R* is calculated by *R* = 1 − (*A* + *T*), where *A* and *T* are absorbance and transmittance, respectively. The calculated average refractive index of different samples is shown in Figure 8.

The average value of the refractive index of undoped ZnO is 2.065, close to the theoretical value [61]. For 1% Mn-ZnO and ZnO thin films co-doped with 1% La and 1% Mn, the average refractive indexes increased to 2.209 and 2.218, respectively. This is due to an increase in transmission. For 1.5% La co-doped in 1% Mn-ZnO film, the refractive index decreased to 2.178. As co-dopants, Mn and La improved ‘*n*’, and with the increase in La concentration, *n* first increased and then decreased. The increase in ‘*n*’ is attributed to increased polarizability [62]. Due to the resonance effect, the electrons are coupled to the oscillating electric field in the ZnO film [26]. Due to this high refractive index, light scattering occurs, which is very suitable for enhancing the lifetime and efficiency of DSSCs. With increasing concentrations of La, ‘*n*’ decreased. The reduction in the refractive index value with increasing La concentration shows that the optical surface dispersion, polarizability, and optical properties of ZnO film were greatly affected. Finally, the efficiency of DSSC was reduced [26]. The formation of the mixed phase of ZnO may be one reason behind this. 

#### 3.2.5. Extinction Coefficient (*k*)

The extinction coefficient (*k*) is a measurement of how well a substance absorbs light at a specific wavelength per unit of mass density. The extinction coefficient ‘*k*’ is determined by the relation $k=\frac{\alpha \lambda }{4\pi }$ and is presented in Figure 9. It is noted that ‘*k*’ increases at shorter wavelengths, decreases at intermediate wavelengths, and increases at longer wavelengths. The existence of absorption caused by interband transitions causes this decrease. This is attributed to the electronic transition to the impurity levels. In addition, the extinction coefficient slightly changed with different doping concentrations. The maximum value of ‘*k*’ was obtained when 0.5% La was doped into 1% Mn-ZnO thin film due to the low *E_g_* value. Because an increase in photon energy also produces an increase in absorption, an increase in the k value will improve DSSC efficiency [26].

#### 3.2.6. Dielectric Constants (*ε_r_* and *ε_i_*) 

The dielectric constant is defined as *ε* = *ε**_r_* + *i**ε**_i_*, where *ε**_r_* is the real part and *ε**_i_* is the imaginary part.

These are calculated by the following formulas.
(13)$$\varepsilon_{r}=n^{2}-k^{2}$$
(14)$$\varepsilon_{i}=2nk$$

Figure 10a,b shows the dielectric constants (a) *ε**_r_* and (b) *ε**_i_* of undoped and co-doped samples. It is observed that the ZnO films co-doped with 0.5% La and 1% Mn have a large range of *ε**_r_* and *ε**_i_*, which is better for solar cell applications. According to Equations (13) and (14), the dielectric constant directly depends upon the values of ‘*k*’ and ‘*n*’. The position density is related to *ε**_r_* and *ε**_i_* in the optical band gap. Photon energy is used by both the imaginary and real parts. Because the strong interactions between photons and electrons take place there, the dielectric constant has a noteworthy impact on the performance of the cell [26].

#### 3.2.7. Conduction and Valence Band Energy

The conduction band (CD) edge of the electron transport layer also plays a vital role in improving the efficiency of solar cells. This edge was calculated by the following method [63]. ZnO has one Zn atom and one oxygen atom. The electron affinities of Zn and O are 0 kJ/mol and −141 kJ/mol, respectively. Similarly, the first ionization energies of Zn and O are 906.4 kJ/mol and 1313.9 kJ/mol, respectively. Both energies are 96.48 kJ/mol, equal to 1 eV. The average of these energies is then calculated.
X_ZnO_ = (4.70 × 6.08)^1/2^ = 5.34
Conduction band energy = E_CB_ = 5.34 − 4.5 − Eg/2 = −0.92 eV

This value is very close to that in the literature [64]. The values were similarly calculated for doped samples and are tabulated in Table 2. The (1%Mn + 1%La)-ZnO film has a low CBE value. Therefore, there is a high driving force for the injection of electrons between the LUMO of the dye and the CB of ZnO. This improves the efficiency of the cell.

### 3.3. DSSC Parameter Measurement

In Figure 11, the voltage and current density plot is shown, which is used to analyze the photovoltaic properties of different DSSCs. From the recorded J-V curve, the short-circuit current density (*J_sc_*) and open-circuit voltage (*V_oc_*) were calculated. The values of these parameters are shown in Table 3. The efficiency (*η*) and fill factor (*FF*) for thin films of DSSCs are calculated by the following formulas [65].
(15)$$FF=\frac{I_{m}\times V_{m}}{J_{SC}\times V_{OC}}$$
(16)$$\eta =\frac{P_{out}}{P_{in}}=\frac{J_{SC}\times V_{OC}\times FF}{P_{in}}$$

The pure ZnO-based DSSC has *V_oc_*, *J_s_*_c_, *FF*, and *η* values of 0.56 V, 2.5 mA/cm^2^, 0.4902, and 0.6864%, respectively. When 1% Mn was doped into ZnO, all photovoltaic values increased, resulting in a value of ƞ of 0.8473%. This efficiency is an increase of 16% compared to the pure ZnO-based DSSC. When a rare-earth metal, i.e., 0.5% and 1% La, is doped into 1% Mn-ZnO, there is an improvement in *V_oc_* and *J_sc_*. Due to an increase in *V_oc_* and *J_sc_*, the efficiency increases by 1.176% with 0.5% La doping and 1.89% with 1% La doping. This increase in efficiency results from the solid contact between the dye and doped Mn or La ions. Ünlü, B. et al. reported that this strong interaction is the result of two factors: first, there is an electrostatic interaction between dye molecules and ions, and then there is direct binding between dye molecules and dopant ions [66]. The maximum *J_sc_* at 1% La doping indicates the large injection efficiency of the cell and a strong interaction between the dye and Mn or La atoms. The maximum FF shows a minimum recombination rate at this doping concentration [67].

For DSSCs based on ZnO co-doped with 1.5% La and 1% Mn, the efficiency decreases to 1.74%, as the electron transport mechanism is confined by a limited electron diffusion process. Therefore, doping produces trapped states that impede the electron movement in DSSCs. These trapped states play a dual role, sometimes increasing the efficiency and sometimes decreasing the efficiency of the device [68]. However, all dopant photoanodes have trapped states; therefore, the number of trapped states increases when increasing the number of dopants. This increase in the number of trapped states might cause recombination and decrease the DSSC efficiency. Hence, La- and Mn-doped ZnO can be used to harvest solar energy, which can be used in photocatalytic applications.

## 4. Conclusions

Pure ZnO thin films and ZnO thin films co-doped with 1% Mn and 0%, 0.5%, 1%, and 1.5% La were successfully fabricated by the sol–gel technique. Pristine ZnO was found to have a hexagonal structure. Mixed hexagonal and cubic phases appeared at co-doping concentrations of 1% Mn and 1.5% La. The high number of La atoms cannot be accommodated in the interstitial spaces but disperse and arrange into a cubic structure in ZnO, causing a material phase change. The grain size was increased by doping, and lattice parameters are remarkably close to the theoretical value. UV–vis results showed that *E_g_* decreased with doping. *E_g_* decreases because of the sp–d exchange interactions between the band and localized electrons. The ‘*n*’ of the undoped ZnO film was 2.065. The maximum ‘*n*’ was obtained for ZnO film co-doped with 1% Mn and 1% La. Therefore, this film has maximum light scattering and is suitable for application in solar cells. The DSSC fabricated with ZnO film co-doped with 1% Mn and 1% La had 1.89% efficiency. This efficiency is 174% higher than that of the pristine ZnO-based DSSC.

## Figures and Tables

**Figure 1 nanomaterials-12-01057-f001:**
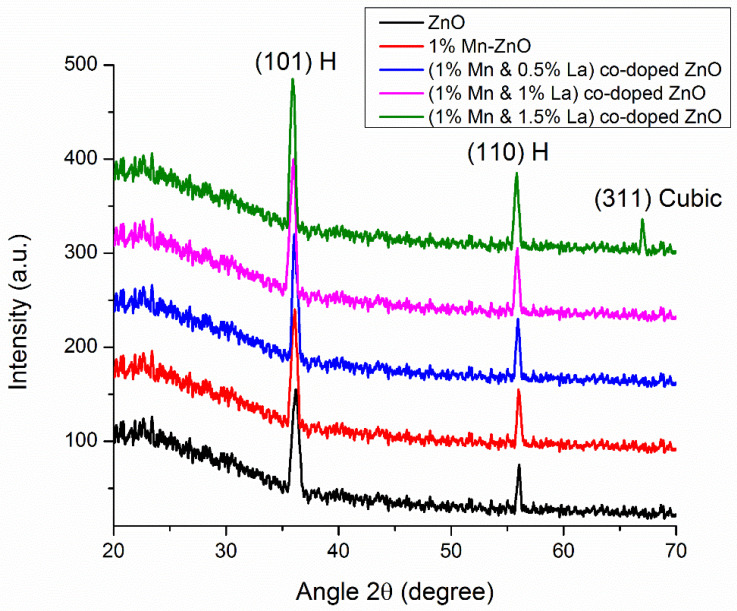
XRD pattern of undoped ZnO thin films and ZnO thin films co-doped with Mn and La.

**Figure 2 nanomaterials-12-01057-f002:**
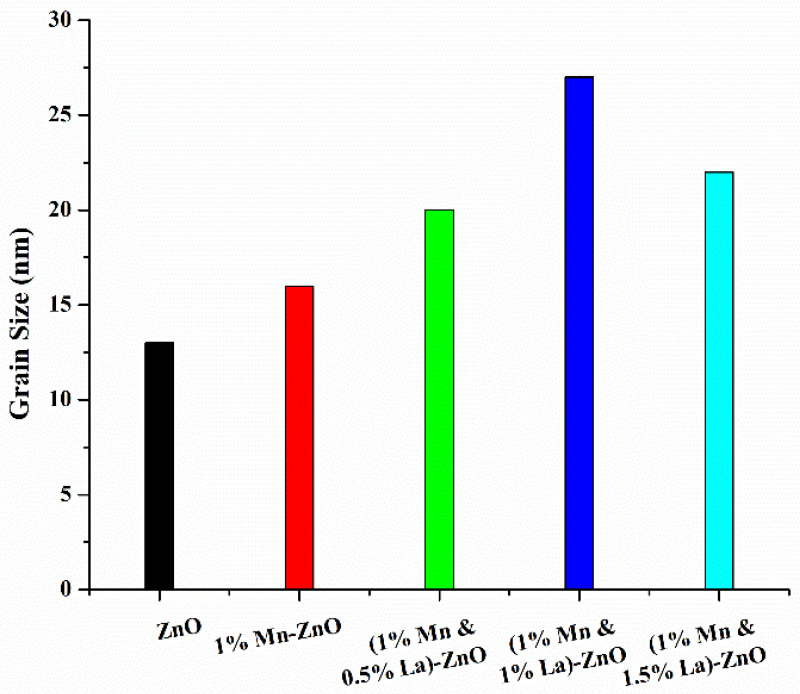
Graph of grain size based on the composition of thin films.

**Figure 3 nanomaterials-12-01057-f003:**
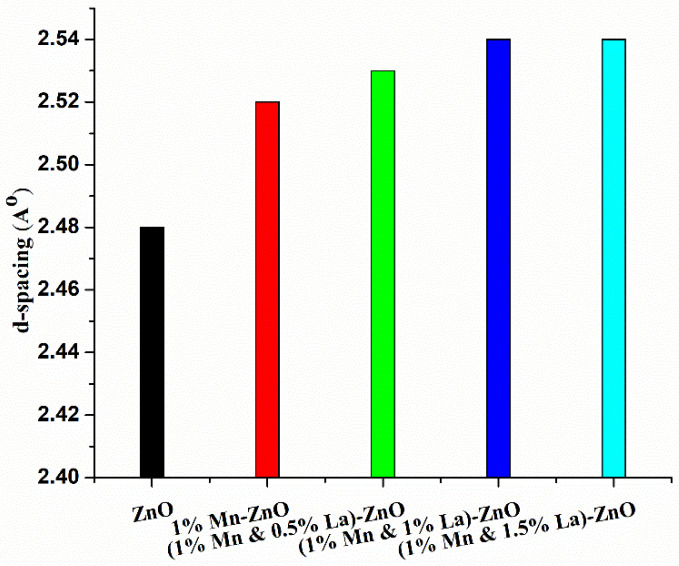
Graph of d-spacing based on the composition of thin films.

**Figure 4 nanomaterials-12-01057-f004:**
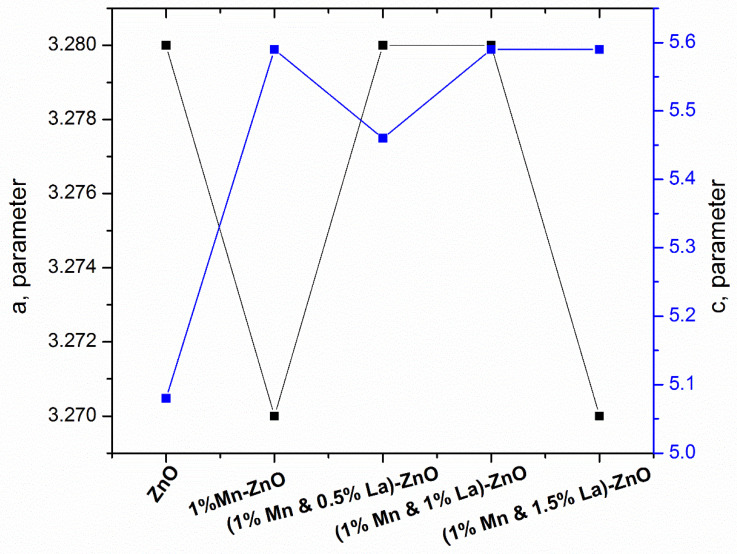
Graph lattice parameters versus pure and doped ZnO film.

**Figure 5 nanomaterials-12-01057-f005:**
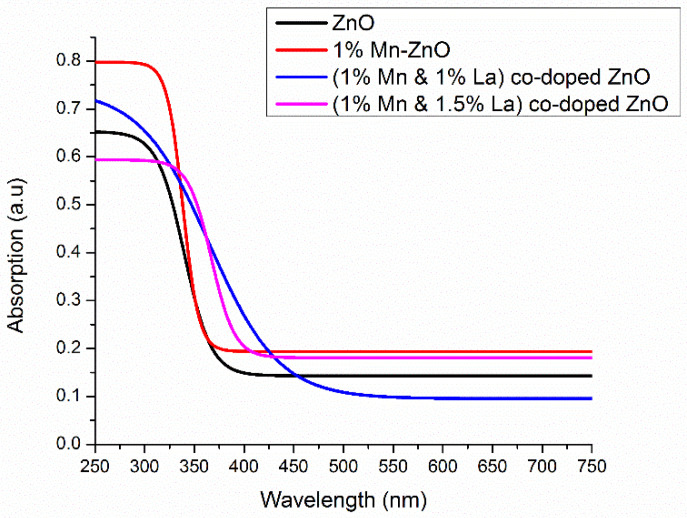
Graph of wavelength versus absorption of films.

**Figure 6 nanomaterials-12-01057-f006:**
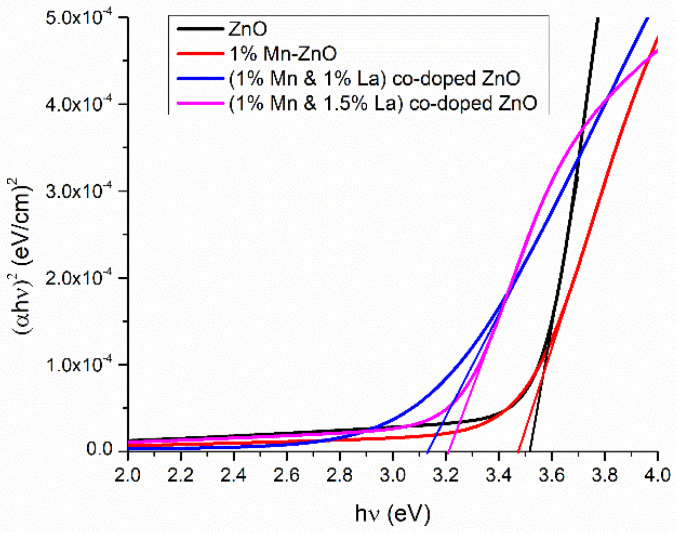
Band gap energy of pure and ZnO thin films co-doped with Mn and La.

**Figure 7 nanomaterials-12-01057-f007:**
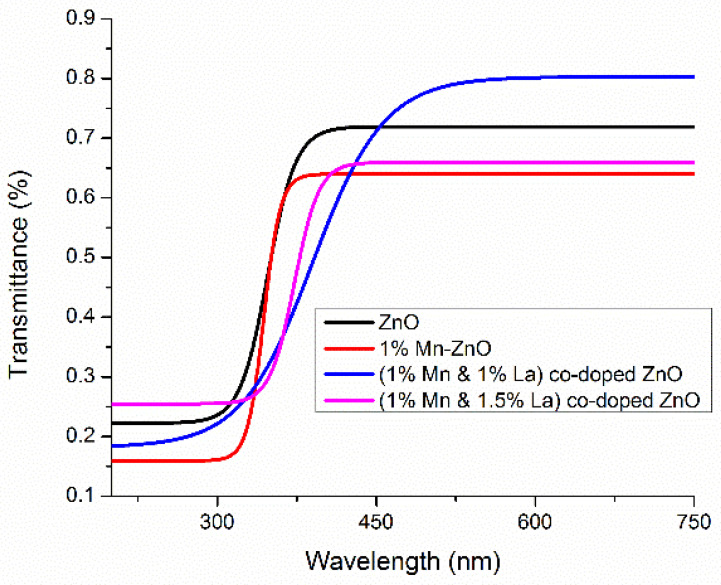
Transmittance of undoped ZnO films and ZnO films co-doped with Mn and La.

**Figure 8 nanomaterials-12-01057-f008:**
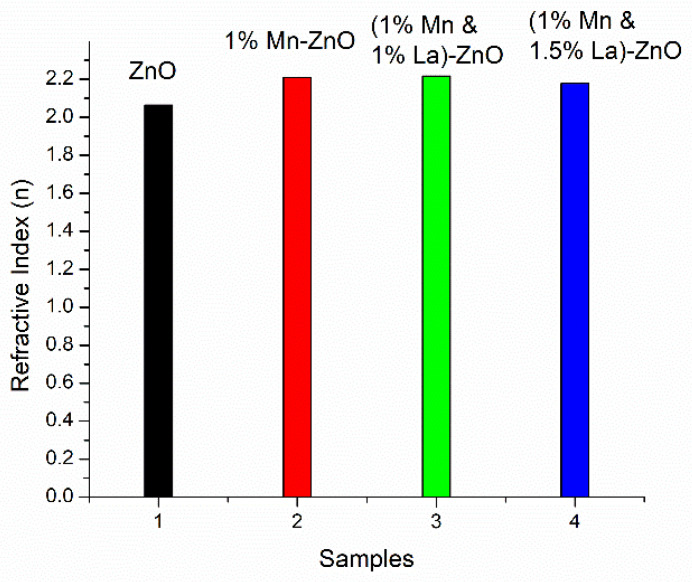
Refractive index of undoped ZnO thin films and ZnO thin films co-doped with Mn and La.

**Figure 9 nanomaterials-12-01057-f009:**
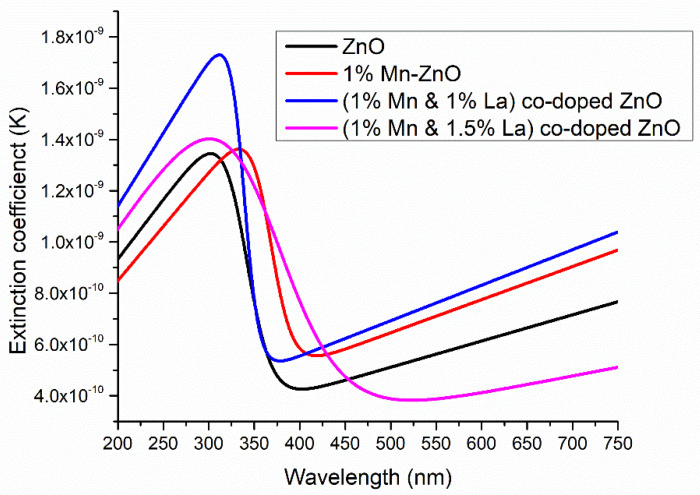
Extinction coefficient of undoped and doped ZnO thin films.

**Figure 10 nanomaterials-12-01057-f010:**
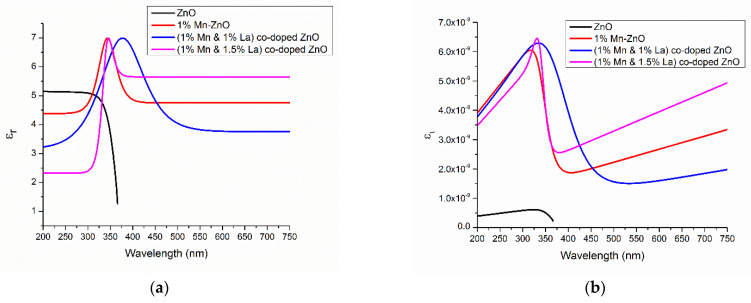
Dielectric constants of (**a**) *ε**_r_* and (**b**) *ε**_i_* of different samples.

**Figure 11 nanomaterials-12-01057-f011:**
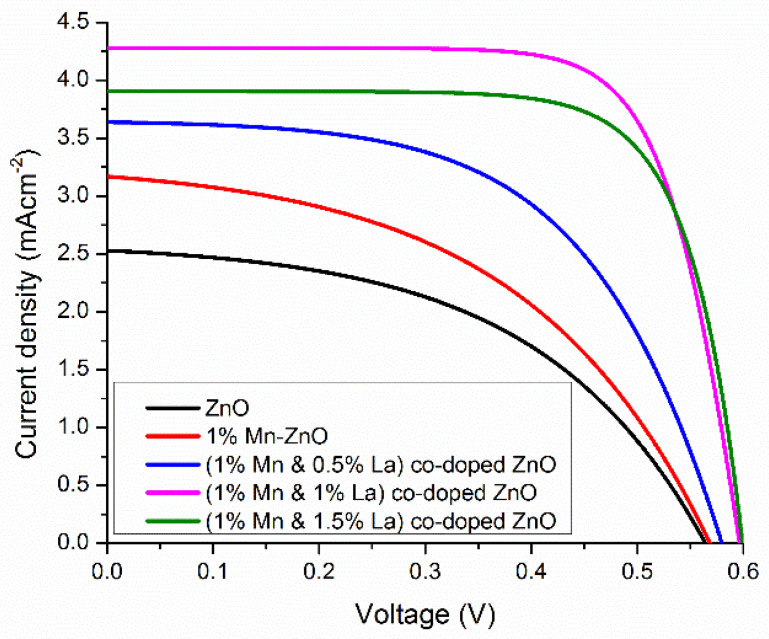
Current density–voltage curves for undoped and co-doped ZnO-based DSSCs.

**Table 1 nanomaterials-12-01057-t001:** Summary of XRD data analysis.

Sr. No.	2θ (Degree)	Grain Size (D) (nm)	Volume of Unit Cell Å^3^	Bond Length (L) (Å)	Lattice Distortion (R) (Å)	d-Spacing (Å)	Atomic Packing Fraction (APF)%	Lattice Constant (Å) a = b, c	Strain (ε)	Crystallinity (Xc)	Phase
Pure ZnO											
1	36.18	13.25	47.27	1.97	0.61	2.483	78.12	a = 3.28 c = 5.08	0.008806	20.86	Hexagonal
2	56.04	21.88	-	-	-	1.641	-		0.003526	32.0	Hexagonal
1%Mn-ZnO											
1	36.11	14.37	51.70	2.03	0.55	2.488	70.78	a = 3.27 c = 5.59	0.001834	22.64	Hexagonal
2	55.99	23.11	-	-	-	1.643	-		0.003341	33.80	Hexagonal
(1%Mn + 0.5%La)-ZnO											
1	36.06	16.40	50.81	2.02	0.56	2.491	72.68	a = 3.28 c = 5.46	0.007621	26.08	Hexagonal
2	55.93	25.24	-	-	-	1.644	-		0.001792	36.92	Hexagonal
(1%Mn + 1%La)-ZnO											
1	36.00	16.58	52.02	2.03	0.55	2.495	70.99	a = 3.28 c = 5.59	0.007097	24.24	Hexagonal
2	55.88	43.16	-	-	-	1.646	-		0.003824	63.15	Hexagonal
(1%Mn + 1.5%La)-ZnO											
1	35.93	15.39	51.70	2.03	0.55	2.500	70.78	a = 3.27 c = 5.59	0.00707	26.08	Hexagonal
2	55.83	20.24	-	-	-	1.647	-		0.003063	29.62	Hexagonal
3	66.99	32.78	-	-	-	1.397	-		0.002004	45.28	Cubic

**Table 2 nanomaterials-12-01057-t002:** Conduction band edges of pure and co-doped ZnO films.

Sr. No.	Samples	Conduction Band Edge
1	ZnO	−0.92
2	1%Mn-ZnO	−0.89
3	(1%Mn + 1%La)-ZnO	−0.72
4	(1%Mn + 1.5%La)-ZnO	−0.76

**Table 3 nanomaterials-12-01057-t003:** DSSC parameter measurements.

Sr. No.	Samples	I_max_ (mA)	V_max_ (V)	J_sc_ (mAcm^−2^)	V_oc_(v)	FF	(η)%
1	ZnO	1.76	0.39	2.5	0.56	0.4902	0.6864
2	1%Mn-ZnO	2.29	0.37	3.17	0.57	0.4689	0.8473
3	(1%Mn + 0.5%La)-ZnO	2.94	0.40	3.63	0.58	0.5585	1.176
4	(1%Mn + 1.0%La)-ZnO	3.93	0.48	4.28	0.59	0.7470	1.8864
5	(1%Mn + 1.5%La)-ZnO	3.55	0.49	3.9	0.598	0.7458	1.7395

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
