# Peer review of "Photovoltaic Properties of ZnO Films Co-Doped with Mn and La to Enhance Solar Cell Efficiency"

_nanomaterials, 2022, doi:10.3390/nano12071057_

Round 1

Reviewer 1 Report

In this paper, the authors investigated the ZnO films co-doped with Mn and La synthesized by sol-gel technique. Structural properties and optical parameters have been calculated.

For this paper, the answer is “Major Revision”. Please modify:

  • At Section 2. Experimentation, please specify how minutes the solution 1 was stirred.
  • Specify the name for FTO.
  • At 3.1.1, specify the reference for “Debye-Scherer’s formula”
  • At 3.2.1, specify the reference for “Bragg’s law”
  • At 3.2.7, line 297, the transmittance spectra of various films are not represented in Figure 1. This is a mistake.
  • At 3.2.5, Figure 9 is not mentioned. Please introduce it.
  • English improvement is required. Some examples but not all are as the following:

Page 6, line 199, modify “3.28nm and the value of parameter ‘c’ is 5.08nm” with “3.28 nm and the value of parameter ‘c’ is 5.08 nm”

Page 10, line 331

Page 11, line 371, modify “doping concentration[55]” with “doping concentration [55].”

Author Response

Reviewer 1 Comments

In this paper, the authors investigated the ZnO films co-doped with Mn and La synthesized by
sol-gel technique. Structural properties and optical parameters have been calculated.
For this paper, the answer is “Major Revision”. Please modify:
- At Section 2. Experimentation, please specify how minutes the solution 1 was stirred.

Ans: Time has been added and highlighted in the manuscript.

- Specify the name for FTO.

Ans: It is fluorene doped tin oxide (FTO). It has been added in the manuscript and highlighted.

- At 3.1.1, specify the reference for “Debye-Scherer’s formula”

Ans: References has been added and highlighted in the manuscript.

- At 3.2.1, specify the reference for “Bragg’s law”

Ans: References has been added and highlighted in the manuscript.

- At 3.2.7, line 297, the transmittance spectra of various films are not represented in Figure
1. This is a mistake.

Ans: It is in Figure 7. It has been corrected and highlighted in the manuscript.

- At 3.2.5, Figure 9 is not mentioned. Please introduce it.

Ans: Figure 9 has been mentioned and highlighted in the manuscript.

- English improvement is required. Some examples but not all are as the following:
Page 6, line 199, modify “3.28nm and the value of parameter ‘c’ is 5.08nm” with “3.28
nm and the value of parameter ‘c’ is 5.08 nm”
Page 10, line 331
Page 11, line 371, modify “doping concentration[55]” with “doping concentration [55].”

Ans: The whole manuscript has been revised, and all the typo mistakes have removed

Reviewer 2 Report

The manuscript describes the preparation of ZnO films, co-doped with Mn and La, as a semiconductor in a photoanode for DSSCs. The authors report
the characterization of the material through a crystallographic analysis of
the film using the XRD technique. In this way, they measured the differences
in crystallographic information depending on the Mn and La used in the
preparation of the film. Furthermore, through UV-Vis spectroscopic analysis
they determine absorption (and hence the bandgap) and transparency. Finally, they build some DSSC cells verifying the increase in efficiency of the cells according to the concentration of Mn and La in the ZnO-film. 
While the crystallographic characterization can be considered sufficient, the measurements made on the DSSC cells are somewhat elementary. To interpret the results (Jsc, Voc and efficiency) it would be necessary to perform impedance measurements. Otherwise, all the hypotheses made by the authors are speculations.  So, in my opinion, the manuscript can be accepted after major revisions: please report impedance measurements for built cells.

Minor revision

In the abstract is reported a Voc value of the best cell equal to 1.89V, while it is about 0.6V (table 2). Please correct. In the abstract, there are some obvious statements like "Sun is the abundant natural resource of solar energy", and the references cited are not appropriate (review about solar energy conversion are better). Careful correction of the text that has grammatical errors and typos must be done.

Author Response

Reviewer 2

Comments and Suggestions for Authors

The manuscript describes the preparation of ZnO films, co-doped with Mn and La, as a semiconductor in a photoanode for DSSCs. The authors report
the characterization of the material through a crystallographic analysis of
the film using the XRD technique. In this way, they measured the differences
in crystallographic information depending on the Mn and La used in the
preparation of the film. Furthermore, through UV-Vis spectroscopic analysis
they determine absorption (and hence the bandgap) and transparency. Finally, they build some DSSC cells verifying the increase in efficiency of the cells according to the concentration of Mn and La in the ZnO-film. While the crystallographic characterization can be considered sufficient, the measurements made on the DSSC cells are somewhat elementary. To interpret the results (Jsc, Voc and efficiency) it would be necessary to perform impedance measurements. Otherwise, all the hypotheses made by the authors are speculations.  So, in my opinion, the manuscript can be accepted after major revisions: please report impedance measurements for built cells.

Response:

Unfortunately, impedance spectroscopy is not available in our university. Impedance spectroscopy is used to determine the internal resistance of cell which affects on the recombination rate. In solar cell parameters, the fill factor tells us about the recombination rate. If fill factor is high then the recombination rate will low and if fill factor is low then the recombination rate will high. In our case 1%Mn and 1% La co-doped ZnO solar cells has high fill factor then it means that it will have low recombination rate and low the internal resistance of cell.  

Minor revision

In the abstract is reported a Voc value of the best cell equal to 1.89V, while it is about 0.6V (table 2). Please correct. In the abstract, there are some obvious statements like "Sun is the abundant natural resource of solar energy", and the references cited are not appropriate (review about solar energy conversion are better). Careful correction of the text that has grammatical errors and typos must be done.

Response:

Voc value in the manuscript has been corrected. The manuscript has been revised, and removed all the spelling and typos mistakes.

Round 2

Reviewer 1 Report

The manuscript was improved. 

At the Section "2. Experimentation", for all reagents, include the manufacturer, city and country.

Author Response

Dear Editor and reviewer

Thank you for valuable suggestions and after the incorporation of suggestions, the Ms is much improved. Point by point explanation of changes, is given below

Comments and Suggestions for Authors

The manuscript was improved. 

At the Section "2. Experimentation", for all reagents, include the manufacturer, city and country.

Response: manufacturer name is provided

Reviewer 2 Report

The manuscript still presents some critical issues:

  1. The phrase "Sun is an abundant natural resource of solar energy" is wrong! Can you tell me what other source of solar energy exists?  The sun is the only source of solar energy! Perhaps the authors wanted to say that the sun is the most abundant source of renewable energy.
  2. References 1 and 2 are not relevant; they must be replaced with this review (https://doi.org/10.1002/chem.201503580)
  3. References 3-5 are not relevant: replace with this review https://doi.org/10.1002/er.4282 .
  4. Add to the references on the DSSC also these:1) 1016/j.rser.2020.109703; 2) 10.3389/fchem.2020.00214.
  5. In the Experimentation section, indicate whether the electrolyte is commercial or, if prepared, report concentration of iodine and iodide and the additives (if presents);
  6. In the paragraph Experimentation in addition to the solar simulator indicate the instrument used for the measurement J-V (source meter).

The authors, through absorbance measurements, calculate the bandgaps of the new materials prepared. However, the energy of the CB is not calculated. Reducing the band gaps could lower the energy of the CB and increase the energy difference between the LUMO of the dye and the CB of the semiconductor, thus increasing the driving force of the electron injection process into the semiconductor. The increase in efficiency of the cells built with the new materials could also be due to this aspect and not only to a better interaction of the dye with the semiconductor due to the presence of Mn or La. Report, if is possible, a table with CB values.

Author Response

Dear Editor and reviewer

Thank you for valuable suggestions and after the incorporation of suggestions, the Ms is much improved. Point by point explanation of changes, is given below

Comments and Suggestions for Authors

The manuscript still presents some critical issues:

  1. The phrase "Sun is an abundant natural resource of solar energy" is wrong! Can you tell me what other source of solar energy exists?  The sun is the only source of solar energy! Perhaps the authors wanted to say that the sun is the most abundant source of renewable energy.

Answer: This sentence has been corrected according to your nice suggestion. Thank you very much.

  1. References 1 and 2 are not relevant; they must be replaced with this review (https://doi.org/10.1002/chem.201503580)

Answer: References 1 and 2 have been removed and (https://doi.org/10.1002/chem.201503580) reference has been added.

  1. References 3-5 are not relevant: replace with this review https://doi.org/10.1002/er.4282.

Answer: References 3-5 have been removed and (https://doi.org/10.1002/er.4282) reference has been added.

  1. Add to the references on the DSSC also these:1) 1016/j.rser.2020.109703; 2) 10.3389/fchem.2020.00214.

Answer: Suggested references have been added in the manuscript.

  1. In the Experimentation section, indicate whether the electrolyte is commercial or, if prepared, report concentration of iodine and iodide and the additives (if presents);

Answer: The electrolyte is commercial purchased from Sigma Aldrich.

  1. In the paragraph Experimentation in addition to the solar simulator indicate the instrument used for the measurement J-V (source meter).

Answer: The source meter is keithley 2400. This has been added and highlighted in the manuscript.